# Using UK Biobank data to establish population-specific atlases from whole body MRI
Sophie Starck [1,5] ✉, Vasiliki Sideri-Lampretsa [1,5] ✉, Jessica J. M. Ritter [2], Veronika A. Zimmer[1],
Rickmer Braren [1,2,3], Tamara T. Mueller [1,5] & Daniel Rueckert [1,4,5]

## Abstract

**Background** Reliable reference data in medical imaging is largely unavailable. Developing tools that allow for the comparison of individual patient data to reference data has a high potential to improve diagnostic imaging. Population atlases are a commonly used tool in medical imaging to facilitate this. Constructing such atlases becomes particularly challenging when working with highly heterogeneous datasets, such as whole-body images, which contain significant anatomical variations.

**Method** In this work, we propose a pipeline for generating a standardised whole-body atlas for a highly heterogeneous population by partitioning the population into anatomically meaningful subgroups. Using magnetic resonance images from the UK Biobank dataset, we create six whole-body atlases representing a healthy population average. We furthermore unbias them, and this way obtain a realistic representation of the population. In addition to the anatomical atlases, we generate probabilistic atlases that capture the distributions of abdominal fat (visceral and subcutaneous) and five abdominal organs across the population (liver, spleen, pancreas, left and right kidneys).

**Results** Our pipeline effectively generates high-quality, realistic whole-body atlases with clinical applicability. The probabilistic atlases show differences in fat distribution between subjects with medical conditions such as diabetes and cardiovascular diseases and healthy subjects in the atlas space.

**Conclusions** With this work, we make the constructed anatomical and label atlases publically available, with the expectation that they will support medical research involving whole-body MR images.

## Plain language summary

Medical imaging requires examples of healthy images to be available for comparison with individual patient data. This comparison is important to detect changes that are indicative of disease and enable diagnosis. Population atlases consist of healthy images that can be used for comparisons. The images should match population characteristics as much as possible. However, building these atlases can be difficult, especially if the images used to compile the atlas show differences. In this study, we provide a method to create a standardised whole-body atlas using whole-body (neck to knee) magnetic resonance images. We produce a set of atlases that represent a healthy population. These atlases have been made publicly available and should assist medical researchers and improve healthcare outcomes for patients.

Magnetic resonance imaging (MRI) is a powerful non-invasive imaging technique that can aid the diagnosis and monitoring of diseases along with treatment planning[1]. These medical images build a valuable basis for medical research, especially if they are available for large cohorts (such as the UK Biobank[2]). They allow for an investigation of inter-subject differences, anomalies, or the comparison of different populations.

Despite all the benefits of large medical imaging cohorts, they come with an important challenge: a lack of comparability between individual subjects, which can, for example, stem from the utilisation of different scanners or different protocols. However, even unavoidable natural anatomical differences between subjects complicate their comparison. One solution to this problem is the introduction of medical atlases. They define a common reference space for all images and represent an average image derived from the whole cohort. To this end, all images of a cohort are registered into a common reference space, enabling a morphological and functional comparison across subjects (inter-subject), groups of subjects (population analysis), or even between the same subject over time (long-itudinal analysis)[3].

[1]Artificial Intelligence in Healthcare and Medicine, School of Computation, Information and Technology, Technical University of Munich, Munich, Germany.
[2]Institute of Diagnostic and Interventional Radiology, Technical University of Munich, School of Medicine, Munich, Germany. [3]German Cancer Consortium (DKTK), Munich partner site, Heidelberg, Germany. [4]Department of Computing, Imperial College London, London, UK. [5]These authors contributed equally: Sophie Starck, Vasiliki Sideri-Lampretsa, Tamara T. Mueller, Daniel Rueckert. ✉e-mail: sophie.starck@tum.de; vasiliki.sideri-lampretsa@tum.de

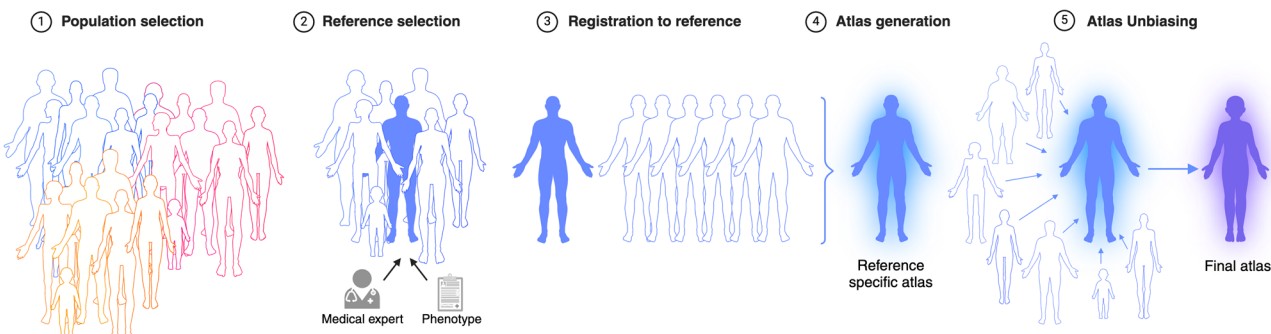

**Fig. 1 | Overview of the processing pipeline for generating the whole-body atlases for each group.** An overview of the processing pipeline for generating the whole-body atlases for each group. The first step is to register all images $I_i$ to the reference, then all images $I_i$ and labels $L_i$ are warped with the resulting deformation fields $\phi_i$ and aggregated in an initial atlas $A_{init}$, the atlases are subsequently unbiased $A_u$ in order to create the final anatomical and label atlases.

Medical atlases have shown great potential to improve research and medical assessment. They can be used to detect anomalies; for instance, by comparing an individual image to the average representation of a population (the atlas), discrepancies can be identified[4] and this can support disease detection[5]. So far, most medical atlases are generated and applied in the field of neuroimaging[5].

There has been a surge in the number of human brain atlas projects, driven by well-funded initiatives such as The BRAIN Initiative[6], The Human Brain Project[7], and The Human Connectome Project[8], to only name a few. These projects aim to advance neuroscience discovery, decode the human brain, study brain circuits and behaviour, map gene expression, obtain ultra-high resolution neuroimages, simulate neocortical micro-circuitry, and understand the brain and its disorders. As a result, the study and the understanding of the human brain are dynamically changing over time as new acquisition techniques, sophisticated applications, tools, and novel concepts are developed[5].

Despite their success in brain imaging, there has been little focus on constructing atlases for other body parts. One reason is that the registration of e.g. whole-body images between different subjects and longitudinally within the same subject is particularly challenging, given the high variability between images. This stems from greatly heterogeneous body compositions throughout the population, and motion- or filling-dependent differences in organ position and shape, as well as motion-dependent artefacts. In contrast, brain images maintain high structural comparability, even if subjects vary in age, sex, height, or weight, whole-body images expose these differences more clearly. Furthermore, there is a lack of whole-body MR image datasets in particular from healthy subjects, due to the time- and cost-intensive scan procedures. This is targeted by large-scale population-wide studies such as the UK Biobank[2] or the German National Cohort (NAKO)[9].

There are, however, a few works investigating the generation and application of whole-body atlases. Still, they have yet to become as popular in analysing these images as their brain counterpart. A potential reason for this could be the fact that most of the currently available whole-body atlases come with major limitations that bound their generalisability: they are often limited in the number of utilised samples, or based on very homogeneous datasets, that only represent a small subset of the whole population. This strongly limits their generalisability to a larger population. Hofmann et al.[10], for example, shows the usage of whole-body atlases for Positron Emission Tomography (PET) attenuation correction, and Karlsson et al.[11] for muscle volume quantification. However, both works use only about 10 subjects for atlas construction. Sjholm et al.[4] propose a multimodal whole-body atlas of functional 18F-fluorodeoxyglucose (FDG) PET and anatomical fat-water MR data and show its utility for anomaly detection. This atlas is constructed from only 27 subjects (15 female, 12 male). An atlas based on a larger dataset of 128 MRI scans from the POEM database (68 female, 60 male) was proposed by Strand et al.[12]. However, this study focuses on analysing only 50-year-old subjects, resulting in an atlas of a very limited subset of the population. The authors study pathophysiological links between obesity, vascular dysfunction, and future cardiovascular disorders, forming a weight loss and gastric bypass study. Using a similar amount of data (250 subjects), Joensson et al.[13] propose an image registration method of whole-body PET-CT images to quantify spatial tumour distributions and capture tissue and muscle mass loss from pre- and post-therapy images. Lind et al.[14] propose to visualise how fat and lean tissue mass is associated with local tissue volume and fat content by utilising whole-body quantitative water-fat MRI and DXA scans of 159 men and 167 women aged 50 in the population-based POEM study. The dataset restriction to a homogeneous distribution for the construction of whole-body atlases emphasises both the complexity of the task, as well as the need for more work in this area.

Given these limitations and the potential of atlas-based research, we investigate the generation of large-scale whole-body atlases on MR images. Our contributions are the following: We introduce (a) a pipeline that allows for generating whole-body, unbiased atlases, considering the high heterogeneity of whole-body MR images. We propose (b) partitioning the whole population into six distinct physiological groups based on sex and body mass index (BMI). For each group, we generate (c) an unbiased anatomical atlas and label atlases for abdominal subcutaneous and visceral fatty tissue and five abdominal organs. Finally, we demonstrate (d) the applicability of these atlases for detecting population subgroup differences between healthy and diseased subjects.

We aim to facilitate further research by making these large-scale whole-body atlases publicly available[15].

## Methods

In this section, we give an overview of the utilised dataset and summarise the applied methods, including the selection criteria for the images, the registration pipeline, and the generation of the atlases, followed by a final unbiasing step. An overview of the atlas generation pipeline is visualised in Fig. 1. The main steps of our atlas generation pipeline are (a) selecting the reference images, (b) registering all healthy MR images and labels to them, (c) averaging the registrations to obtain the anatomical and label atlases, and (d) unbiasing the generated atlases to make them more representative. More details about the individual steps follow in this section.

### Dataset

The atlases provided in this work are constructed from the whole-body MR images from the UK Biobank dataset, a large-scale dataset containing genetic, lifestyle, health data and biological measurements from half a million subjects[2]. It is an ongoing longitudinal study performed in the UK and represents a selection across the population in the UK with an age range of the subjects between 40 and 80 years. Additionally, it contains a large selection of phenotypic and lifestyle information, including imaging data types such as brain, heart, or whole-body MR at different time points. The UK Biobank currently provides ca. 50, 000 T1-weighted, dual-echo gradient whole-body MR images with a size of [224 × 168 × 363] voxels and a resolution of [2, 23 × 3 × 2, 23] mm. They contain 4 Dixon contrasts: water,

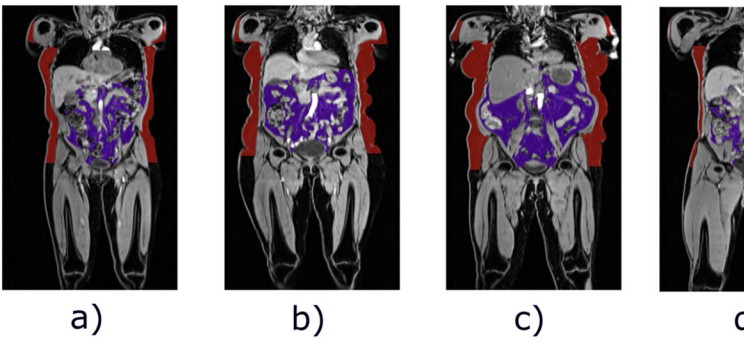
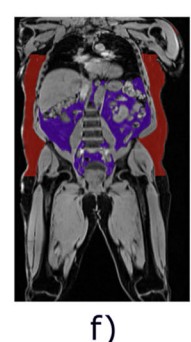

**Fig. 2 | Reference images of the different subgroups of the dataset.** Reference images of the different subgroups of the dataset. The first three images show the female subjects, **a–c** which are, respectively, the normal weight, overweight, and obese groups. The normal weight, overweight and obese male reference images are shown in **d–f**. All images are overlayed with their respective labels of fatty tissue. Red: abdominal subcutaneous fat, purple: abdominal visceral fat.

fat, in-phase and out-of-phase. These scans were acquired in six individual stations and, in a post-processing step, merged using a public stitching tool[16]. This way, one image from neck to knee is generated for each subject. This study utilises 3000 of the stitched water-contrast images; the selection process is described in the following paragraph and example images are shown in Fig. 2. We extract segmentations for abdominal organs and abdominal fat, following the pipelines proposed in Kart et al.[17] and Küstner et al.[18]. The organ segmentations are extracted for the liver, the spleen, the pancreas, and the left and the right kidney. In addition, the fat segmentations are divided into abdominal subcutaneous and visceral fatty tissue. We note that the fat segmentation algorithm only targets abdominal fat identification, not considering fatty tissue in other body parts, such as the legs. Additionally, the data distribution of the UK Biobank, in terms of ethnicity, is highly unbalanced towards white British subjects. We, therefore, note that our atlases contain the same inherent imbalance. The UK Biobank has ethical approval from the North West Multi-centre Research Ethics Committee to handle human participant data, no additional ethical approval was required because the study involved the secondary use of data. Written informed consent was obtained from all participants and all data is deidentified for analysis. Eligible researchers may access UK Biobank data on www.ukbiobank.ac.uk upon registration. For this study, permission to access and analyse the UK Biobank data was approved under the application 87802.

**Data selection.** Given the high intra-subject physiological variability, we consider one single atlas insufficient to represent the entire cohort. Thus, we separate the dataset into six groups based on BMI and sex. We generate one anatomical atlas and corresponding label atlases for each of the following subject groups: (1) females with normal BMI, (2) overweight females, (3) obese females, (4) males with normal BMI, (5) overweight males, and (6) obese males. Table 1 indicates the specific BMI ranges for each group. The categorisation utilised in this work functions as a medically motivated example; BMI and sex were chosen to differentiate groups as a coarse anatomical boundary, but our pipeline can be used for arbitrary subgroupings (e.g. body composition, age).

For the generation of each atlas, to generate a representation of a healthy population, we only select subjects that have no record of cancer, no self-reported diseases, and no operation history in each category (matching BMI and sex). We select all available subjects that meet those criteria; all numbers are reported in Table 1.

**Atlas generation pipeline**

In the following, we introduce our atlas generation pipeline, visualised in Fig. 1. The first step of generating an atlas requires the registration of all images to align the whole dataset in one common coordinate space. This is done by choosing a *reference* image, also called *fixed*, and registering all remaining *moving* images of the dataset to it (Fig. 1 Section 1-3). By applying the resulting transformations to the images and the corresponding segmentations, we generate one initial anatomical atlas and several initial label

**Table 1 | Overview of the different BMI categories**

| Category | BMI range | # females | # males |
|---|---|---|---|
| Overweight | [25; 30] | 1071 | 1363 |
| Obese | ≥30 | 612 | 555 |

These categories were used to split the data for the atlas generation and the amount of data used per group.

**Table 2 | Reference characteristics**

| Sex | BMI Category | Age (years) | Weight (kg) | Height (cm) | BMI | Body fat (%) |
|---|---|---|---|---|---|---|
| | Overweight | 64 | 71.2 | 163.0 | 26.8 | 42.2 |
| | Obese | 61 | 87.6 | 162.0 | 33.4 | 42.0 |
| Male | Normal weight | 66 | 72.9 | 177.0 | 23.3 | 18.5 |
| | Overweight | 68 | 83.1 | 176.0 | 26.8 | 26.3 |
| | Obese | 67 | 101.6 | 174.0 | 33.6 | 31.9 |

Body composition values of the selected reference subjects for each BMI group and sex.

atlases. Finally, all atlases are unbiased to avoid strongly aligning with the reference image.

**Reference selection.** For each subgroup of the dataset, one reference image is selected. This reference image can highly impact the registration performance and must be carefully selected to optimally represent the distribution in the cohort. We use the phenotypic data of the subjects to identify the most representative reference for each group. All reference subjects represent the median age, weight, height, BMI, and body fat percentage in the respective subgroup of the dataset (e.g. overweight females). The specific properties of all selected reference subjects are summarised in Table 2. A medical expert then assessed the selected references, evaluating whether the muscle and fat proportions were representative of the respective subgroup. Figure 2 shows the final reference images selected for the atlas generations, overlayed with the corresponding abdominal fat segmentations. We can see apparent differences in anatomy and fat distribution between the groups, highlighting the necessity to separate the dataset.

**Pre-processing.** Although all images are acquired using the same scanner and following the same procedure, imaging data subgroup variability remains high primarily for high inter-subject variation in fat distribution and organ size, shape and localization, raising challenges for subsequent registration. To mitigate this and subsequently maximise registration performance, we apply two types of normalisation: intensity-based and spatial normalisation. We first perform an intensity-based normalisation in several steps. (1) We ensure the same range of pixel values in all images by min-max normalisation. (2) We enhance the

contrast of the images by thresholding the intensity histogram to accentuate boundaries and, therefore, facilitate registration, and (3) mask out the background of the images using an automatically generated body mask in order to avoid any unwanted impact of the background noise on the registration. Given the substantial anatomical differences between subjects (e.g. height), the dataset shows large variability regarding angles and fields of view. To address this, we perform a spatial normalisation using a centre of mass initialisation. Within each subgroup, the centre mass of each subject is computed and aligned to the corresponding reference's centre of mass, with an iterative closest point (ICP) method using the open-source library by[19]. This alignment is a global registration initialisation to correct extreme positional differences and aims to facilitate the following registration steps.

**Registration.** To construct an atlas, i.e. an average representation of the data, we must ensure that all the data lies in the same coordinate system. In other words, we have to ensure a spatial normalisation of the data that accounts for spatial misalignment due to how the data was acquired. Registration is the key step in this process, ensuring a spatial normalisation and mapping all images into the same coordinate space[3]. The goal of registration is to find the optimal transformation that minimises the differences between corresponding features in the two images. In particular, given a fixed $\mathcal{F}: \Omega_F \subseteq \mathbb{R}^n \to \mathbb{R}$ and moving image $\mathcal{M}: \Omega_M \subseteq \mathbb{R} \to \mathbb{R}^n$, image registration aims to find the best spatial transformation $\phi^*: \Omega_F \to \Omega_M$ that minimises a cost function or error metric for a pair of images. This can be formalised as:

$$\phi^* = \arg\min_{\phi}[\mathscr{D}(\mathscr{F}, \mathscr{M} \circ \phi) + \lambda \mathscr{R}_{Reg}], \tag{1}$$

where $\mathcal{D}$ is a dissimilarity metric between the fixed ($\mathcal{F}$) and the transformed moving image ($\mathcal{M} \circ \phi$), warped in the coordinate space of the fixed using the transformation $\phi^*$, $\mathcal{R}_{Reg}$ is a regularisation term on the transformation which is furthermore introduced to enforce smoothness and diffeomorphism weighted by a factor $\lambda$.

Whole-body image registration is a non-trivial task due to major anatomical and physiological variations in organ position and shape. Two major sources of variation are (1) the breathing cycle-dependent craniocaudal positioning of the diaphragm, which can cause several centimetres of organ displacement and deformation of primarily upper abdominal organs and (2) differences in the filling state of the oesophagus, stomach, bowel, gallbladder and bladder. In contrast to the highly deformable regions, there are also rigid structures such as the bones or the spinal cord.

Considering these, we construct the atlases by performing registration in two steps in this work. First, we perform a global registration step, affine registration, that ensures that two bodies are coarsely aligned. This allows us to account for large misalignments, e.g. due to the position of the subjects in the scanner. In the second step, we use the affinely registered images and perform deformable registration to account for non-rigid, local discrepancies, e.g. breathing motion and distribution of subcutaneous fat. In the two following paragraphs, we provide more details about the two steps, as well as the software that we used and the choices we made.

**Affine Registration** As a first step, we register all images using *affine* registration, i.e., compensate for differences in the *translation*, *rotation*, *shearing* and *scaling* of the images. This registration step aims to bring all the subjects to roughly the same space. It accounts for large global misalignment, e.g., the subject's position in the scanner and spacing and orientation discrepancies. However, if we constructed the atlases compensating only for affine transformations, this would lead to very blurry and low-quality atlases that would not be very useful for any downstream task. For this reason, we perform deformable transformation as a second step to alleviate local non-linear differences between images.

**Deformable Registration** Subsequently, we use the affinely registered images as the starting point to perform *deformable* registration, also known

as non-linear registration. This type of registration accounts for more complex, spatially varying transformations between points or local features, e.g., organ deformations, breathing motion, and fat distribution.

To construct the atlases, we want to ensure that the computed deformable transformations are invertible and topology-preserving. As a result, we choose to model them as diffeomorphic transformations. In other words, instead of computing the displacements for every point in space, we compute the velocities and integrate them[20] to obtain the desired displacements[21]. Moreover, since an accurate deformable transformation is very challenging to achieve, especially in the abdominal region, due to its versatility, we choose to model it using Free-Form deformation (FFD)[22]. Using this technique, we estimate the transformation on a lattice of regularly spaced control points and interpolate its values to obtain the final displacement field over the whole domain. The spacing of the control points affects the transformation smoothness, with smaller spacing allowing for more complex transformations. However, small spacing might also be more prone to spatial folding, i.e., implausible spatial configurations. As a result, careful tuning is required to ensure a good balance between these competing properties. Last, but not least, to accurately recover large deformable transformations, we decided to incorporate a multi-resolution technique in registering the images. The intuition is that we first perform an alignment at a coarse level and refine the registration until we reach the finest resolution level. To do this, we repetitively downsample the images by a factor of two. Then, we start the process by registering the coarsest level. Using the estimated transformation as initialisation for the next, finer level, we repeat the process until we reach the finest level. The multi-resolution technique allows us to use a top-down approach to recover the larger deformable misalignments, avoiding getting stuck to local minima.

**Atlas generation and unbiasing.** Once the subjects from each group are registered to their corresponding reference image and consequently to the same coordinate space, we construct an initial atlas $A_{init}$ for each group by averaging the $n$ registered subjects of the group:

$$A_{init} = \frac{1}{n}\sum_{i=1}^{n} \mathscr{I}_i \circ \phi_i, \tag{2}$$

where $\mathcal{I}_i$ refers to the image of the $i$-th subject in the group and $\phi_i$ to its corresponding transformation field. We follow this approach to generate one initial anatomical atlas and all label-based atlases (for abdominal organs and fat) for each subgroup of the dataset.

These initial atlases heavily rely on the choice of reference. Figure 3 shows how similar the reference-specific atlas (b) is to the reference image (a). This atlas still serves as a decent representation of the population, even with this bias, since the reference image was statistically and clinically chosen to best represent the subgroup of the population. It, however, holds the risk of propagating subject-specific traits to the atlas, which might be mistaken with population traits in further downstream tasks. We introduce an atlas unbiasing[23] step to the pipeline to address this. This process utilises the deformation fields and the previously constructed atlas to generate a more realistic representation of the population. We extract an average inverse transformation field $\Phi$ by first inverting the individual transformation fields and then averaging them:

$$\Phi = \frac{1}{n}\sum_{n=1}^{n} \phi_i^{-1}. \tag{3}$$

The unbiased Atlas $A_u$ is then derived by applying the average inverse transformation field $\Phi$ to the initial atlases:

$$A_u = A_{init} \circ \Phi = A_{init} \circ \left(\frac{1}{n}\sum_{n=1}^{n} \phi_i^{-1}\right). \tag{4}$$

**Fig. 3 | Atlas unbiasing example for the obese female subgroup.** Unbiasing example on the obese female atlas. From left to right: the reference image (**a**), the initial (biased) anatomical atlas (**b**), and the unbiased atlas (**c**).

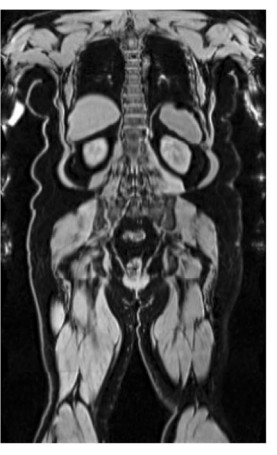
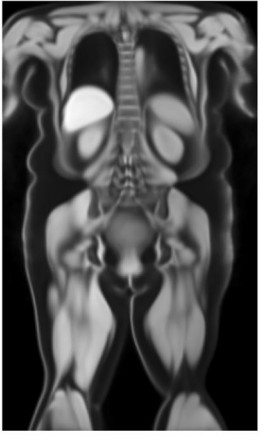
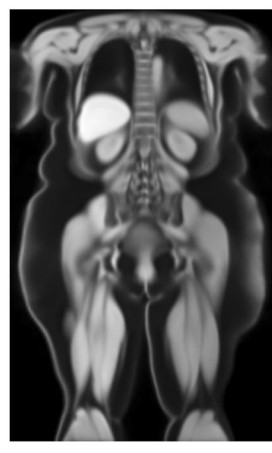

**(a)** Reference image  **(b)** Initial anatomical atlas  **(c)** Unbiased anatomical atlas

**Voxel based morphometry**. The atlases proposed in this work are systematically built to represent a healthy population. We show their applicability to detect anomalies or physiological deviations from the atlas by performing voxel-based morphometry (VBM)[24] using a neuroimaging toolbox[25]. This method is conventionally used to quantify grey or white matter concentration voxel-wise. It allows for the identification of regions of interest and holds the potential for more localised studies of the brain. VBM analysis comprises several steps: first, one or several interest groups are identified, and then the target tissue is segmented and registered for all subjects across each group. Spatial smoothing is applied to account for extra variability after registration; finally, the groups are compared using voxel-wise statistical testing.

Here, we utilise the abdominal visceral fat maps from healthy subjects that were previously used to build the atlas and test them against diseased subjects. Two pathological subgroups are selected, one containing all subjects suffering from CAD and another one from type 2 diabetes. The subjects are subsequently registered to the atlas space, with the same pipeline used to build the atlas. Due to the limited availability of pathological data, we use a subset of the healthy data to create cohorts with comparable sample sizes. Each label map is then spatially normalised with a Gaussian kernel; this allows each voxel to represent the average of itself and its neighbours, making each voxel comparable between the two groups. We then fit a General Linear Model (GLM) to the data and derive the associated statistics via a parametric test (t-test) to identify voxels that significantly explain the variability in visceral fat between the healthy atlas and the pathological group. We obtain a voxel-wise z-score map (z-map), indicating the number of standard deviations away from the mean. Higher values indicate stronger evidence against the null hypothesis, suggesting significant differences in fat volume between the groups. Performing simultaneous statistical tests introduces the multiple comparisons problem, which refers to the increased chance of false positives with many tests. We apply a correction for multiple comparisons called False Discovery rate (FDR) to mitigate this effect, and we only retain highly significant *p*-values $p < 0.001$ from the corrected z-map.

**Statistics and reproducibility**. All data analysis was performed with Python 3.11.3 and is reproducible using the code linked below in Code availability[26]. All the data used was obtained as described in the Data Availability section. For all statistical tests, a *p*-value $< 0.001$ was considered statistically significant.

**Reporting summary**
Further information on research design is available in the Nature Portfolio Reporting Summary linked to this article.

## Results
With this work, we release all six large-scale anatomical atlases for the different population subgroups (based on BMI and sex), as well as the corresponding label atlases of abdominal organs and abdominal fatty tissue. Figure 4 shows an example of these atlases for overweight female subjects. As an effort to facilitate research in the area of large-scale body imaging and population studies, we make these atlases publicly available via CERN's open repository Zenodo, at this address: https://doi.org/10.5281/zenodo.13136891[15].

The following sections describe the experimental setup for this pipeline, the resulting atlases and an example use case of these atlases for anomaly detection between healthy and two distinct pathological groups.

### Experimental setup
Two criteria are important to assess when choosing the most suitable registration algorithm: accuracy and plausibility. To enforce an optimal tradeoff between both criteria, we compare two established registration methods: MIRTK[27] and Voxelmorph[28].

Specifically, we use the open-source toolbox *Deepali*[29], which is a GPU-accelerated implementation of MIRTK[27]. This is a pairwise optimisation approach that allows for an extremely fast hyperparameter search and registration due to its GPU support. The second algorithm we investigate is the widely adopted learning-based method, Voxelmorph[28], based on convolutional neural networks. To obtain the results, we register each group to its reference image. Since we do not have ground truth transformations to assess the registration performance, we do so by reporting surrogate measures for the registration accuracy and regularity. To assess the registration accuracy, we report the DICE score (label overlap) of the liver, spleen, right and left kidney, the mean dice over all labels in one group and the mean Hausdorff distance (95$^{th}$ percentile). We assess the registration regularity by measuring the ratio of spatial foldings. To do this, we compute the Jacobian determinant of the transformation and then calculate the ratio of voxels with negative ones over the total amount of voxels.

Table 3 demonstrates the quantitative results of each method for each BMI group. Regarding accuracy, MIRTK reported consistently better results than its learning-based counterpart, Voxelmorph. This is not the case for registration regularity, where Voxelmorph computes consistently more plausible transformations. To construct atlases that could be useful for downstream tasks, we need to achieve high registration accuracy. We opted for MIRTK as the registration method of choice due to its superior accuracy. Although Voxelmorph exhibited slightly better plausibility scores, MIRTK's higher accuracy ensured a more reliable registration outcome, making it the preferred approach for our application. For more details on the selected hyperparameters, we refer the interested reader to the supplementary material 1.

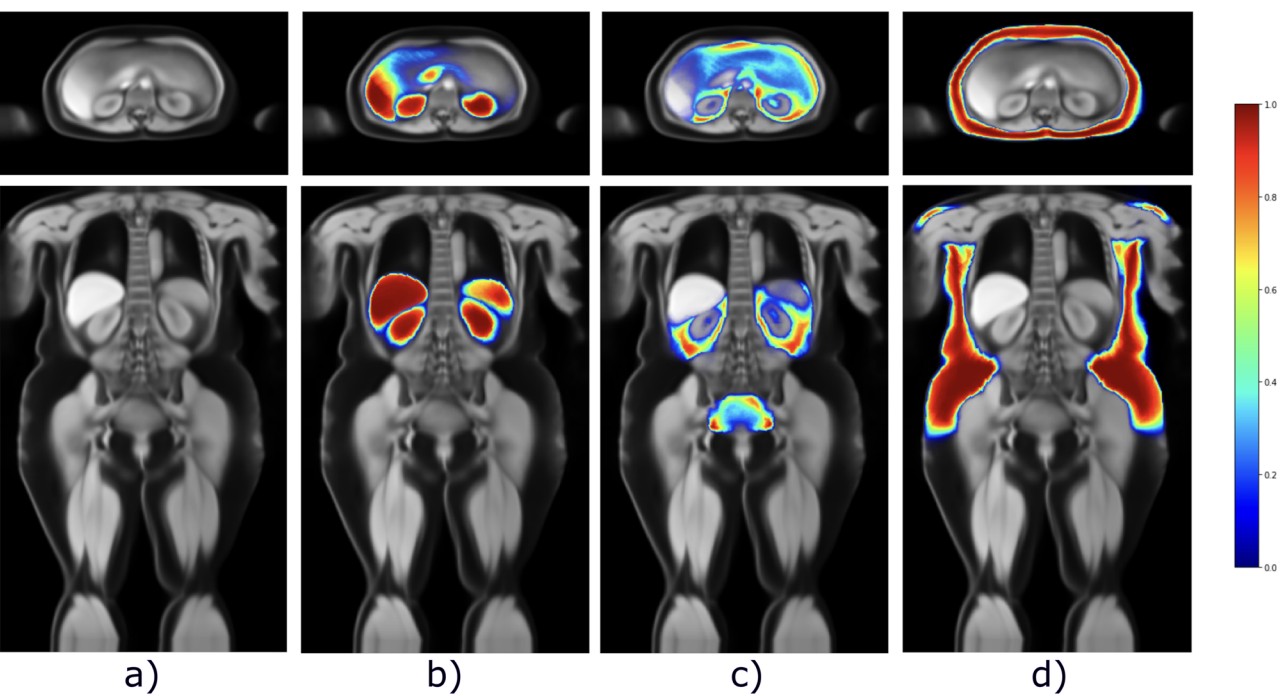

**Fig. 4 | Example atlas of the overweight female subgroup.** We visualise one axial slice and one coronal slice of the unbiased anatomical atlas (**a**), overlayed with the probability maps of the abdominal organs (**b**), abdominal visceral fat (**c**) and the abdominal subcutaneous fat (**d**).

### Table 3 | Quantitative results of different registration methods

| | Method | Liver↑ | Spleen ↑ | Kidney L↑ | Kidney R↑ | Mean Dice↑ | Mean HD 95% ↓ | Folding Ratio ↓ |
|---|---|---|---|---|---|---|---|---|
| **Female** | | | | | | | | |
| **healthy** | Pre-reg | 0.49 ± 0.19 | 0.12 ± 0.13 | 0.15 ± 0.16 | 0.21 ± 0.20 | 0.24 ± 0.14 | 17.92 ± 7.16 | - |
| | Affine | 0.63 ± 0.19 | 0.43 ± 0.16 | 0.55 ± 0.14 | 0.51 ± 0.16 | 0.53 ± 0.07 | 13.35 ± 5.41 | - |
| | VMorph [28] | 0.81 ± 0.10 | 0.58 ± 0.16 | 0.69 ± 0.14 | 0.63 ± 0.18 | 0.67 ± 0.11 | 13.43 ± 6.36 | **0.0081 ± 0.0076** |
| | **MIRTK** [27] | **0.83 ± 0.13** | **0.60 ± 0.16** | **0.70 ± 0.13** | **0.69 ± 0.19** | **0.70 ± 0.08** | **12.40 ± 7.34** | 0.0232 ± 0.0034 |
| **over-weight** | Pre-reg | 0.52 ± 0.15 | 0.26 ± 0.18 | 0.35 ± 0.20 | 0.31 ± 0.20 | 0.36 ± 0.09 | 17.78 ± 5.41 | - |
| | Affine | 0.70 ± 0.09 | 0.48 ± 0.15 | 0.63 ± 0.13 | 0.58 ± 0.15 | 0.60 ± 0.08 | 11.83 ± 3.99 | - |
| | VMorph [28] | 0.84 ± 0.04 | 0.59 ± 0.19 | 0.69 ± 0.15 | 0.64 ± 0.16 | 0.69 ± 0.09 | 10.49 ± 6.38 | **0.0047 ± 0.0042** |
| | **MIRTK** [27] | **0.85 ± 0.07** | **0.62 ± 0.14** | **0.78 ± 0.08** | **0.73 ± 0.15** | **0.74 ± 0.08** | **9.59 ± 4.38** | 0.0303 ± 0.0036 |
| **obese** | Pre-reg | 0.47 ± 0.19 | 0.20 ± 0.21 | 0.20 ± 0.21 | 0.22 ± 0.22 | 0.27 ± 0.11 | 28.39 ± 9.66 | - |
| | Affine | 0.65 ± 0.07 | 0.36 ± 0.14 | 0.41 ± 0.17 | 0.48 ± 0.18 | 0.48 ± 0.11 | 22.90 ± 6.90 | - |
| | VMorph [28] | **0.84 ± 0.06** | **0.58 ± 0.20** | 0.62 ± 0.16 | 0.65 ± 0.18 | 0.67 ± 0.10 | 10.08 ± 8.83 | **0.0051 ± 0.0044** |
| | **MIRTK** [27] | 0.78 ± 0.07 | **0.58 ± 0.15** | **0.73 ± 0.17** | **0.71 ± 0.16** | **0.70 ± 0.07** | **9.57 ± 6.70** | 0.0436 ± 0.0073 |
| **Male** | | | | | | | | |
| **healthy** | Pre-reg | 0.46 ± 0.21 | 0.26 ± 0.20 | 0.29 ± 0.20 | 0.27 ± 0.21 | 0.32 ± 0.08 | 18.49 ± 8.22 | - |
| | Affine | 0.72 ± 0.13 | 0.50 ± 0.16 | 0.58 ± 0.16 | 0.59 ± 0.18 | 0.60 ± 0.08 | 10.02 ± 5.61 | - |
| | VMorph [28] | 0.84 ± 0.14 | 0.62 ± 0.20 | 0.62 ± 0.23 | 0.65 ± 0.24 | 0.68 ± 0.17 | 10.28 ± 6.68 | **0.0088 ± 0.0021** |
| | **MIRTK** [27] | **0.88 ± 0.14** | **0.64 ± 0.18** | **0.70 ± 0.18** | **0.75 ± 0.17** | **0.74 ± 0.08** | **6.61 ± 6.27** | 0.0297 ± 0.0047 |
| **over-weight** | Pre-reg | 0.46 ± 0.19 | 0.16 ± 0.17 | 0.20 ± 0.18 | 0.21 ± 0.18 | 0.30 ± 0.09 | 19.66 ± 7.87 | - |
| | Affine | 0.71 ± 0.08 | 0.46 ± 0.16 | 0.54 ± 0.14 | 0.52 ± 0.16 | 0.56 ± 0.09 | 11.26 ± 4.68 | - |
| | VMorph [28] | 0.84 ± 0.05 | 0.64 ± 0.18 | 0.72 ± 0.19 | 0.73 ± 0.14 | 0.73 ± 0.10 | 11.32 ± 6.91 | **0.0044 ± 0.0047** |
| | **MIRTK** [27] | **0.85 ± 0.09** | **0.67 ± 0.17** | **0.75 ± 0.12** | **0.74 ± 0.14** | **0.75 ± 0.06** | **10.59 ± 6.58** | 0.0303 ± 0.0041 |
| **obese** | Pre-reg | 0.45 ± 0.14 | 0.20 ± 0.16 | 0.16 ± 0.15 | 0.19 ± 0.16 | 0.25 ± 0.11 | 22.80 ± 7.05 | - |
| | Affine | 0.65 ± 0.13 | 0.46 ± 0.18 | 0.50 ± 0.17 | 0.47 ± 0.18 | 0.52 ± 0.23 | 21.79 ± 15.47 | - |
| | VMorph [28] | 0.76 ± 0.19 | 0.57 ± 0.22 | 0.60 ± 0.27 | 0.61 ± 0.24 | 0.63 ± 0.19 | 16.32 ± 14.91 | **0.0011 ± 0.0012** |
| | **MIRTK** [27] | **0.81 ± 0.15** | **0.66 ± 0.18** | **0.71 ± 0.20** | **0.62 ± 0.21** | **0.70 ± 0.07** | **15.43 ± 16.12** | 0.0364 ± 0.0081 |

We register every image for the group's reference. Then, to assess the registration accuracy, we report the DICE scores for 4 for different organs and mean dice score and the 95th percentile of the Hausdorff distance over all labels, as well as the folding ratio for the transformation regularity. The best performance for each metric is highlighted in bold.

## Anatomical and label atlases

The atlas generation pipeline described above is performed for each different subgroup. This results in several atlases for each group: (1) an anatomical atlas showing the average anatomy, (2) two fatty tissue atlases for abdominal subcutaneous and visceral fat, and (3) one atlas for each abdominal organ (liver, spleen, pancreas, left and right kidney).

The anatomical atlas allows for a general analysis of the whole body regarding overall shape and structure. In contrast, the label atlases (organs and fat) allow for a more focused study of the distribution of specific structures in the body. Figure 4 shows an example of these atlases (a), overlaying the organ (b), the visceral (c) and the subcutaneous fat (d) atlases on the anatomical atlas for the female, overweight subgroup. The label atlases can be interpreted as probability maps, indicating the likelihood of finding a certain structure (e.g., the liver) at a specific position in the atlas. A more extensive visualisation of all atlases is available in the Supplementary Figs. 2–4.

We visualise the results of the atlas unbiasing in Fig. 3. It shows the reference image (a), the initial (biased) anatomical atlas (b) and the unbiased anatomical atlas (c) of the female obese BMI subgroup. The unbiased anatomical atlas (c) is still crisp but contains fewer anatomical specificities propagated by the reference. The reference image (a) and the initial atlas (b) are visually very similar, which is especially obvious when looking at the body outline or the liver shape. In contrast, the unbiased atlas (c) is smoother in these areas and not as strongly conditioned on the reference image. We

can also see that the unbiased atlas presents more subcutaneous fat than the initial atlas in the hip area, which occurs when most of the population in the subgroup has more fat than the reference in this region. The unbiasing step allows us to capture this deviation from the reference and, therefore, correct the atlas. This shows how crucial this step is in order to obtain representative atlases for the whole population. More results of the unbiasing step on all groups are visible in Supplementary Fig. 1.

The UKBB provides whole-body MR scans of four different contrasts: water, fat, in-phase, and out-of-phase. So far, all anatomical atlases have been shown using the water-contrast images since we performed the atlas generation on those images. However, the other contrast images can easily be registered to the atlas using the existing corresponding transformations, resulting in atlases for all contrast images, e.g., fat-contrast anatomical atlas.

## Anomaly detection in anatomical atlases

We showcase the clinical value of these atlases of the generated atlases by presenting a use case that employs the label atlases. We demonstrate that the atlases can be successfully used to detect changes in distribution for diseased subgroups, more specifically, coronary artery disease (CAD) and type 2 diabetes. We do so by performing Voxel-Based Morphometry (VBM), a statistical approach performing a voxel-wise comparison between two groups, in this case, a healthy group and a pathological one. Figure 5 shows the resulting statistical maps overlaid on the anatomical atlas for the obese female subgroup. Both sections show the significant voxels highlighted by

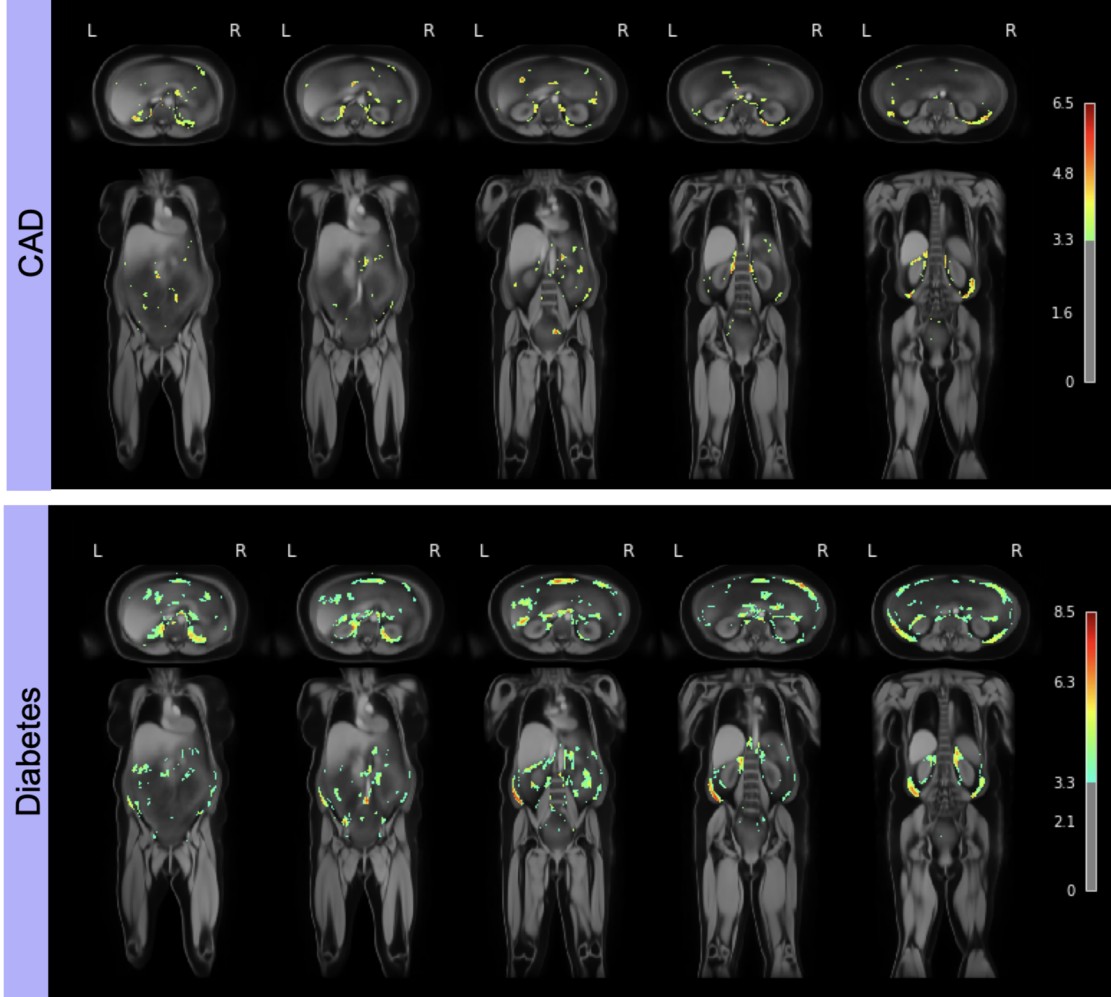

**Fig. 5 | Significant voxels in the visceral fatty tissue for patients with coronary artery disease and type 2 diabetes.** Significant voxels ($p < 0.001$) in the visceral fatty tissue for patients with coronary artery disease and type 2 diabetes for the female obese subgroup. The columns show different slices for both axial and coronal views. This shows the existence of significantly more visceral fat in the CAD ($n = 312$) and diabetic group ($n = 393$) than in the healthy one. The colourbar describes the z score; higher values indicate significant differences in visceral fat between the groups.

the VBM overlayed on the anatomical atlas. The first section shows the test for the healthy subgroup against CAD and the second against type 2 diabetes. Each column visualises a different slice for the axial (first row) and the coronal views (second row). A more comprehensive set of slices can be visualised in the Supplementary Figs. 5–8.

We see a significant increase in retroperitoneal fat around the kidneys, as well as in extraperitoneal fat within the lower pelvis (Fig. 5).

Whereas the CAD group displays a selective increase in the perirenal visceral fat compartment, a more global increase is noted in the Type 2 diabetes group (Fig. 5). This finding is of particular interest because an increase in the perirenal visceral fat compartment has previously been noted as an early indicator of atherosclerosis and CAD development[30,31], supporting the validity of employing label atlases in anomaly detection. Moreover, we also observe increased fat concentrations around the pancreas for the diabetic group (Fig. 5), which the atrophy of the organ could explain. Type 2 diabetes is directly correlated with pancreatic function[32], a significant change in the region compared to the healthy group is therefore highly plausible.

## Discussion

In medical imaging, reliable reference data is, for the most part, still missing. This poses an important limitation of diagnostic imaging tests. Medical atlases serve as a strong candidate for reference and are a powerful tool for spatial normalisation and population-wide analyses of medical images. They have become a staple in neuroimaging research and are now commonly used in most brain analysis pipelines such as VBM[24]. However, atlases have been widely under-explored on other medical images, particularly more global representations, such as whole-body scans. The rise of large-scale whole-body datasets such as the UK Biobank[2] or the German National Cohort[9] opens new possibilities in this area. We make use of the data-rich UK Biobank and generate several large-scale atlases for whole-body MR images and provide a systematic pipeline for atlas generation and population analysis with whole-body images (Fig. 1). One challenge of whole-body images is the extremely high inter-subject variability, which is due to the large differences in anatomy, e.g., based on height, sex, age, or BMI. Considering this, we split the dataset into six distinct subgroups by sex and BMI (Table 1) and generate different atlases for each subgroup. Furthermore, we only selected subjects without any self-reported diseases and no record of cancer in order to obtain a general representation of a healthy population.

For each BMI and sex group, we generate one anatomical atlas, which is based on the water-contrasted whole-body MR images, along with several label atlases, utilising segmentations for five abdominal organs (liver, spleen, pancreas, left and right kidney), as well as visceral and abdominal subcutaneous fatty tissue. These label atlases provide a probabilistic interpretation of the likelihood of a structure of interest (e.g., an organ) to be located at a specific position in the anatomical atlas. They can be used to identify subjects or groups that deviate from the atlas, i.e., anomaly detection, and investigate distributional differences between subgroups, for example, by assessing how the fat distribution varies. We showcase this by performing VBM between the atlas and groups of subjects with coronary artery diseases and type 2 diabetes. With this, we are able to expose variation in distribution between groups at a voxel level. This holds great potential for population studies and group-wise analysis, as it allows for spatial quantification of change. The substantial medical benefit results from the non-invasive, radiation-free measurement of, for example, body composition.

These atlases aim to serve as a representative for each group, describing coarse anatomical properties, which renders them unsuited by nature for high-frequency analysis. Parenchymatous organs and more rigid structures in the body are more easily represented than softer structures in the atlas due to their high inter-subject variability. The gut region is an example where the variability is so high that the registration performance is suboptimal, and the averaging step smooths out all relevant structures. This renders the atlases inadequate for tasks such as intestinal health analysis. The registration step

was performed by optimising global similarity-based parameters. A limitation of this optimisation scheme is the tradeoff between highly accurate registration and spatial folding (unrealistic deformation). To obtain plausible deformations, one has to sacrifice registration accuracy, which is the leading cause of resulting blurry regions in the atlas. A way to ensure that regions with high variability are accurately registered would be to introduce an adaptive regularisation for the registration[33–35] based on specific regions. This would allow the introduction of medically informed registration, where more rigid regions could be more heavily registered. Recent advancements in full-body MR segment anything models[36] would allow targeting very specific regions. Moreover, extracting such labels holds the potential for generating more extensive label maps and extending the possibilities for anomaly detection. Muscle tissue, for example, is a strong indicator of age and frailty[37], which would be valuable to integrate into the atlases provided here, along with other label atlases such as the spine, heart, bone, or lung atlases.

VBM approaches are a standard neuroimaging processing pipeline, which has proven useful for analysing the relationship between brain anatomy and covariates[38,39]. However, the pipeline is very sensitive to design choices[40,41], and the multiple steps associated with the analysis introduce uncertainty, arising from misregistration or remaining false positives from the t-test, for example. This is especially evident in the case of a more challenging dataset such as full-body MR. We, for instance, attribute the highlighted disconnected patches in the gut region in Fig. 5, to noise arising from this.

A potentially valuable application of these atlases would be to leverage them in the context of longitudinal studies. For a subset of the whole dataset, the UK Biobank also provides images acquired at several time points. These could be used to investigate intra-subject developments over time. Furthermore, a comparison between different datasets, such as the NAKO[9], would show the usability of the atlases beyond one single dataset. By making all generated atlases publicly available, we envision supporting medical research on the UK Biobank and whole-body images since we believe whole-body imaging holds great potential for population studies and group sub-typing.

## Data availability

Eligible researchers may access UK Biobank data on www.ukbiobank.ac.uk, upon registration. For this study, permission to access and analyse the UK Biobank data was approved under the application 87802. The overview figure was created in BioRender. Starck, S. (2023) BioRender.com/d53h428. The atlases generated in this work are available at https://doi.org/10.5281/zenodo.13136891[15].

## Code availability

All codes associated with this study are available at https://github.com/starcksophie/WholeBodyAtlas[26].

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

## Acknowledgements
TM and SS were supported by the ERC (Deep4MI - 884622). SS has furthermore been supported by the Federal Ministry of Education and Research (BMBF). RB was supported by the Federal Ministry of Education and Research (BMBF, Grant Nr. 01ZZ2315B and 01KX2021), the Bavarian Cancer Research Centre (BZKF, Lighthouse AI and Bioinformatics) and the German Cancer Consortium (DKTK, Joint Imaging Platform).

## Author contributions
S.S., V.S-L, T.T.M, R.B. and D.R. designed the study and conceived the experiments. S.S. V.S-L. and T.T.M. conducted the implementation and experiments. S.S., V.S-L, T.T.M, D.R. and J.J.M.R. analysed the results. J.J.M.R. and R.B. provided medical interpretation of the results. S.S., V.S-L, and T.T.M. wrote the manuscript. V.A.Z., R.B., T.T.M., and D.R. provided supervision. All authors reviewed the manuscript.

## Funding

## Competing interests
The authors declare no competing interests.
