## [Transparent Peer Review file · Communications Medicine]

Using UK Biobank data to establish population-specific atlases from whole body MRI

Corresponding Author: Ms Sophie Starck

Version 0:

Reviewer comments:

Reviewer #1

(Remarks to the Author)

Key results

The authors propose a pipeline to generate population atlases of abdominal organ and tissue masks from mri data. It involves the alignment of a set of whole body mri scans to a reference scan and subsequently unbias them via deformation fields. The authors differentiate between bmi and sex, resulting in 6 atlases. The resulting data is provided via zenodo. The authors propose potential use cases on what these atlases might be used for.

Validity

The authors provide few quantitative information regarding their experiments. Extra information should be added to the experiments.

No results are shown for non-parenchymatous abdominal organs. Also in the figures, the intestine is not visible. Please add these images to show the limitations. For an application of the atlases, maybe these regions should be masked?

Why is BMI used for stratification and not e.g. body composition parameters?

The resulting structures are rather coarse and anatomical variations are not detectable. Please discuss this point.

As one of the applications of atlases, label propagation is mentioned. The current progress in the field of whole-body segmentations makes this application less important.

Significance

For brain imaging, standard atlases are very important. So far, these atlases have not been extended to the whole body. Many structures outside the brain do show a much higher variability and are strongly influenced by breathing and filling state of the organs. This is a challenging task that has not been solved with the presented approach. Yet, it is a step in this direction, and this paper could be cited very well.

Data and methodology

The method is a combination of multiple existing methods for registration and analysis. The resulting data sets are made available by the authors.

Analytical approach

The method is not compared against other pipelines/baselines.

The qualitative results should be confirmed and discussed, e.g. why do portions of the lung contribute to fatty liver disease?

Maybe utilize VBM (https://www.fil.ion.ucl.ac.uk/spm/doc/papers/john_vbm_methods.pdf) Instead of a voxel-wise- t-test.

Clarity and context

The paper is well written. The limitations should be discussed and future work, e.g. how to address non-parenchymatous organs should be sketched.

On Page 7, within the section anatomical and label atlases, an entire section is repeated from earlier parts of the paper. Please rephrase or remove.

Recommendation

Major review

Reviewer #2

(Remarks to the Author)

Medical imaging atlas on large population databanks is an effective way of modelling the variability in the data, capturing inter-subject differences, and linking them to diseases and outcomes. The authors proposed an atlases pipeline using UK Biobank whole-body MRI materials. They generated atlases for 6 subgroups, using subsets of UKB materials. The atlases are able to show the anomaly features in liver steatosis, CAD and type 2 diabetes subgroups. The availability of the derived datasets is a strength.

The reviewer has some major reservations, regarding, (1) limited novelty in the atlas methods, (2) the data selection approach in utilising large UK Biobank and (3) very limited, preliminary results and statistical analysis.

Specifically,

1. Methods.

The authors used pre-existing affine and non-rigid image registration models/toolboxes, with no reporting of performance on the UKBB dataset.

In combining affine and deformable registration in 'Deepali', the authors described: "We refrain from rerunning this processing step to avoid unnecessarily increasing the environmental impact of this work." This is not a solid justification for not using 'Deepali' in both steps. In fact, the authors may compare different existing methods, reporting and comparing the performance to provide more insights in selecting the best approach.

2. The data selection approach. Selecting 3000 samples for 6 subgroups may not be the best way of using the databank. For example, can the authors use all materials that fits the criteria of each subgroup? No data bias will be introduced as each subgroup is analysed independently.

Similar: "page 8: We randomly select 200 subjects with high liver fat." Why not analyse ALL subjects with high liver fat (e.g. in male overweight group)?

Page 8: "We select 200 subjects with CAD, 200 subjects with diabetes and 200 subjects from the healthy atlas group." The authors should use all subjects that are available and fit the criteria, to utilise the large data size and statistical power of UKBB.

3. Very limited report of results is the reviewer's main concern.

- There are no validation results of the affine and non-rigid registration algorithms on UKBB. The authors should report dice coefficients and bland-altman plots on organ regions of interest (ROI), e.g. (by transforming the ROI masks back to original images, and comparing them with original annotations) to provide more insight into the accuracy, bias and probability of failure/misregistration. This is evidently crucial, as the authors observed diffused noise and significant voxels introduced by mis-registration in Figs 5 and 6.

- Each result subsection only has one demonstrative example for unbiased atlas, with no summary of performance or findings on all samples.

- The results section has a large volume of texts that describe the methods and approaches, and of discussion. They should be moved to the right sections. Subsequently, the results section should report, e.g. new findings, association with outcomes, diagnostic performance of the derived atlas, the performance of label propagation. The authors mentioned label propagation in the discussion, this is straightforward to run and can significantly improve the paper.

Version 1:

Reviewer comments:

Reviewer #1

(Remarks to the Author)

The authors have successfully incorporated the Reviewer's comments and made the work even better. No further concerns.

Reviewer #2

(Remarks to the Author)

The authors have addressed all my previous comments point by point. This is a much more strengthened manuscript and I have no further suggestions.

Reviewer #1 (Remarks to the Author):

We would like to thank Reviewer #1 for providing insightful feedback and highlighting improvement points for our manuscript. This constructive feedback has prompted us to revise and enhance the clarity of our work. We address the individual discussion points in the subsequent rebuttal and have revised the manuscript accordingly.

Key results

The authors propose a pipeline to generate population atlases of abdominal organ and tissue masks from mri data. It involves the alignment of a set of whole body mri scans to a reference scan and subsequently unbias them via deformation fields. The authors differentiate between bmi and sex, resulting in 6 atlases. The resulting data is provided via zenodo. The authors propose potential use cases on what these atlases might be used for.

Validity

The authors provide few quantitative information regarding their experiments. Extra information should be added to the experiments.

We would like to thank the reviewer for pointing this out. Based on their suggestion, we added another section to our manuscript (**Experimental setup**), where we compare the selected pairwise optimisation method (MIRTK) with a broadly used learning-based method (Voxelmorph) and report quantitative results assessing the transformations' accuracy and regularity. More specifically, we report the dice overlap of 4 organs (liver, spleen, left and right kidney), the mean dice and the mean Hausdorff distance (95th percentile). Moreover, we assess the resulting transformation regularity by reporting the folding ratio (Ratio of negative Jacobian determinant values). These results are summarised in Table 3, which contains the quantitative results of comparing the registration methods.

No results are shown for non-parenchymatous abdominal organs. Also in the figures, the intestine is not visible. Please add these images to show the limitations. For an application of the atlases, maybe these regions should be masked?

We would like to thank the reviewer for their comment. Indeed, regions of high variability, e.g. intestines, are very challenging to register accurately. As a result, registering and averaging these regions over a cohort can produce blurry results. We added respective visualisations (Figures 7 and 8) to the supplementary material and a paragraph in the "**Discussion and Conclusion**" section that specifically discusses these limitations of the atlases introduced by registration.

"Parenchymatous organs and more rigid structures in the body are more easily represented than softer structures in the atlas due to their high inter-subject variability. The gut region is an example where the variability is so high that the registration

performance is suboptimal, and the averaging step smooths out all relevant structures. This renders the atlases inadequate for tasks such as intestinal health analysis. The registration step was performed by optimising global similarity-based parameters. A limitation of this optimisation scheme is the tradeoff between highly accurate registration and spatial folding (unrealistic deformation). To obtain plausible deformations, one has to sacrifice registration accuracy, which is the leading cause of resulting blurry regions in the atlas.”

We agree that masking blurry regions would be a valid option to address this limitation. However, we refrain from masking out any regions in the final atlases, as we want to provide the most general atlas possible. Furthermore, the application we are showcasing uses the fatty tissue in the abdominal region, which is adjacent to the gut. Any inaccurate masking in this area might introduce a bias to our application. However, for different use cases, masking the blurry regions in the atlas might be beneficial. Depending on the specific requirements, we leave it to the user to introduce this as a postprocessing step.

Why is BMI used for stratification and not e.g. body composition parameters?

We would like to thank the reviewer for this comment. Indeed, using the BMI to define subpopulations has been a design choice of our method. We agree that body composition parameters could also have been chosen as a valid stratification criterion. In this work, we used BMI and sex as a broad, descriptive, and accessible categorisation. Furthermore, the pipeline we propose in our work is by design highly flexible to these design choices. Therefore, alternative parameters can be selected if one is interested in different subgroups. We added this to the manuscript in the “**Data Selection**” section (page 3) in the following way:

”The categorisation utilised in this work functions as a medically motivated example; BMI and sex were chosen to differentiate groups as a coarse anatomical boundary, but our pipeline can be used for arbitrary subgroupings (e.g. body composition, age).”

The resulting structures are rather coarse and anatomical variations are not detectable. Please discuss this point.

We agree that the anatomical variations within our selected groups (BMI, sex) are merged into one atlas. We would like to highlight that especially in high variable regions, e.g. abdominal region containing intestines and visceral fat, we run into the issue that more accurate registration, which would lead to sharper atlases, creates more spacial folding, affecting the plausibility of the result. Consequently, it was necessary to balance these two competing properties and the visual appearance of the final atlases. We recognise this as a limitation of our work, and added a specific paragraph discussing this issue in the “**Discussion and Conclusion**” section (page 10):

“Parenchymatous organs and more rigid structures in the body are more easily represented than softer structures in the atlas due to their high inter-subject variability. The gut region is an example where the variability is so high that the registration performance is suboptimal, and the averaging step smooths out all relevant structures. This renders the atlases inadequate for tasks such as intestinal health analysis. The registration step was performed by optimising global similarity-based parameters. A limitation of this optimisation scheme is the tradeoff between highly accurate registration and spatial folding (unrealistic deformation). To obtain plausible deformations, one has to sacrifice registration accuracy, which is the leading cause of resulting blurry regions in the atlas. A way to ensure that regions with high variability are accurately registered would be to introduce an adaptive regularisation for the registration [1, 2, 3] based on specific regions.”

[1] Y. Wang, H. Qiu and C. Qin, "Conditional Deformable Image Registration with Spatially-Variant and Adaptive Regularization," *2023 IEEE 20th International Symposium on Biomedical Imaging (ISBI)*, Cartagena, Colombia, 2023, pp. 1-5, doi: 10.1109/ISBI53787.2023.10230464.

[2] Simpson, I. J. et al. Probabilistic non-linear registration with spatially adaptive regularisation. *Med. image analysis* 26, 203–216 (2015).

[3] Simpson, I. J. et al. A bayesian approach for spatially adaptive regularisation in non-rigid registration. In *Medical Image Computing and Computer-Assisted Intervention–MICCAI 2013: 16th International Conference, Nagoya, Japan, September 22-26, 2013, Proceedings, Part II* 16, 10–18 (Springer, 2013)

As one of the applications of atlases, label propagation is mentioned. The current progress in the field of whole-body segmentations makes this application less important.

We thank the reviewer for pointing this out. We agree that recent advances in learning-based segmentation and specifically large models, e.g. Segment anything in medical images [4], Total Segmentator [5], make the segmentation task easier. We still believe that label propagation could be useful in cases where these models fail or when dealing with out-of-distribution data, e.g. different datasets, different modalities.

[4] Ma, Jun, Yuting He, Feifei Li, Lin Han, Chenyu You, and Bo Wang. "Segment anything in medical images." *Nature Communications* 15, no. 1 (2024): 654.

[5] Sebro, R. and Mongan, J., 2023. TotalSegmentator: A Gift to the Biomedical Imaging Community. *Radiology: Artificial Intelligence*, 5(5), p.e230235.

Significance

For brain imaging, standard atlases are very important. So far, these atlases have not been extended to the whole body. Many structures outside the brain do show a much higher variability and are strongly influenced by breathing and filling state of the organs. This is a challenging

task that has not been solved with the presented approach. Yet, it is a step in this direction, and this paper could be cited very well.

Data and methodology

The method is a combination of multiple existing methods for registration and analysis. The resulting data sets are made available by the authors.

Analytical approach

The method is not compared against other pipelines/baselines.

We would like to thank the reviewer for pointing this out to us. In the new version of the manuscript, we have added baselines and quantitative metrics. We kindly refer to our reply above, for a more detailed description.

The qualitative results should be confirmed and discussed, e.g. why do portions of the lung contribute to fatty liver disease?

Regarding fatty liver disease, we have removed this application from the manuscript to have a more in-depth focus on the analysis of diabetes and cardiovascular patients. Subsequently, we have added a more detailed discussion part to our application section, including expert analysis from radiologists, as follows:

“ We see a significant increase in organ-associated visceral fat around the kidneys, as well as in mesenteric fat and within the lower pelvis (Figure 5). Whereas the CAD group displays a selective increase in the perirenal visceral fat compartment, a more global increase is noted in the Type 2 diabetes group (Figure 5, bottom row). This finding is of particular interest because an increase in the perirenal visceral fat compartment has previously been noted as an early indicator of atherosclerosis and CAD development [6, 7], supporting the validity of employing label atlases in anomaly detection. Moreover, we also observe increased fat concentrations around the pancreas for the diabetic group (Figure 5), which the atrophy of the organ could explain. Type 2 diabetes is directly correlated with pancreatic function [8], a significant change in the region compared to the healthy group is therefore highly plausible.”

[6] Wang, W., Lv, F. Y., Tu, M. & Guo, X. L. Perirenal fat thickness contributes to the estimated 10-year risk of cardiovascular disease and atherosclerotic cardiovascular disease in type 2 diabetes mellitus. *Front. Endocrinol.*15, 1434333 (2024).

[7] Okeahialam, B. N., Sirisena, A. I., Ike, E. E. & Chagok, N. M. Ultrasound assessed peri-renal fat: an index of sub-clinical Atherosclerosis. *Am. J. Cardiovasc. Dis.* 10, 564 (2020).

[8] Marshall, S. M. The pancreas in health and in diabetes. *Diabetologia*, 1962–1965 (2020)

Maybe utilize VBM (https://www.fil.ion.ucl.ac.uk/spm/doc/papers/john_vbm_methods.pdf)
Instead of a voxel-wise- t-test.

We would like to note that we are indeed using VBM for our analysis. However, we believe that this was not very clearly stated in the text. In the new version, we further clarify this in the Methods section with a new section (**Voxel Based Morphometry**). We specifically use the neuroimaging library nilearn [9].

[9] Abraham, A. et al. Machine learning for neuroimaging with scikit-learn. *Front. Neuroinformatics* 8, 71792 (2014)

Clarity and context

The paper is well written. The limitations should be discussed and future work, e.g. how to address non-parenchymatous organs should be sketched.

We thank the reviewer for their comment. We have added a more detailed discussion on this matter to the manuscript. We refer the reviewer to the discussion part of the paper:

“Parenchymatous organs and more rigid structures in the body are more easily represented than softer structures in the atlas due to their high inter-subject variability. The gut region is an example where the variability is so high that the registration performance is suboptimal, and the averaging step smooths out all relevant structures. This renders the atlases inadequate for tasks such as intestinal health analysis. The registration step was performed by optimising global similarity-based parameters. A limitation of this optimisation scheme is the tradeoff between highly accurate registration and spatial folding (unrealistic deformation). To obtain plausible deformations, one has to sacrifice registration accuracy, which is the leading cause of resulting blurry regions in the atlas. A way to ensure that regions with high variability are accurately registered would be to introduce an adaptive regularisation for the registration [1, 2, 3] based on specific regions.”

[1] Y. Wang, H. Qiu and C. Qin, "Conditional Deformable Image Registration with Spatially-Variant and Adaptive Regularization," *2023 IEEE 20th International Symposium on Biomedical Imaging (ISBI)*, Cartagena, Colombia, 2023, pp. 1-5, doi: 10.1109/ISBI53787.2023.10230464.

[2] Simpson, I. J. et al. Probabilistic non-linear registration with spatially adaptive regularisation. *Med. image analysis* 26, 203–216 (2015).

[3] Simpson, I. J. et al. A bayesian approach for spatially adaptive regularisation in non-rigid registration. In *Medical Image Computing and Computer-Assisted Intervention–MICCAI 2013: 16th International Conference, Nagoya, Japan, September 22-26, 2013, Proceedings, Part II* 16, 10–18 (Springer, 2013)

On Page 7, within the section anatomical and label atlases, an entire section is repeated from earlier parts of the paper. Please rephrase or remove.

We would like to thank the reviewer for this comment. We made the respective changes in the manuscript.

Reviewer #3 (Remarks to the Author):

We would like to thank Reviewer #3 for providing insightful feedback on our manuscript. Based on their remarks, we believe our manuscript has improved significantly.

Medical imaging atlas on large population databanks is an effective way of modelling the variability in the data, capturing inter-subject differences, and linking them to diseases and outcomes. The authors proposed an atlases pipeline using UK Biobank whole-body MRI materials. They generated atlases for 6 subgroups, using subsets of UKB materials. The atlases are able to show the anomaly features in liver steatosis, CAD and type 2 diabetes subgroups. The availability of the derived datasets is a strength.

The reviewer has some major reservations, regarding, (1) limited novelty in the atlas methods, (2) the data selection approach in utilising large UK Biobank and (3) very limited, preliminary results and statistical analysis.

Specifically,

1. Methods. The authors used pre-existing affine and non-rigid image registration models/toolboxes, with no reporting of performance on the UKBB dataset.

We would like to thank the reviewer for pointing this out. Based on their suggestion, we added another section to our manuscript (**Experimental setup**), where we report the dice overlap of 4 organs (liver, spleen, left and right kidney), the mean dice and the mean Hausdorff distance (95th percentile). We also assess the resulting transformation regularity by reporting the folding ratio (Ratio of negative Jacobian determinant values).

These results are summarised in Table 3, which contains the quantitative results of comparing the registration methods.

In combining affine and deformable registration in 'Deepali', the authors described: "We refrain from rerunning this processing step to avoid unnecessarily increasing the environmental impact of this work." This is not a solid justification for not using 'Deepali' in both steps. In fact, the authors may compare different existing methods, reporting and comparing the performance to provide more insights in selecting the best approach.

In order to address this concern, we re-ran our entire pipeline using the same toolbox (Deepali). We note that our results did not change, but we agree that this is more consistent. Furthermore, we want to stress that Deepali is a GPU-accelerated

implementation of MIRTk, both tools therefore have the same underlying algorithm. We also clarify this in our manuscript (Section Experimental Setup):

“To enforce an optimal tradeoff between both criteria, we compare two established registration methods: MIRTk [10] and Voxelmorph [11]. Specifically, we use the open-source toolbox Deepali [12], which is a GPU-accelerated implementation of MIRTk”

In addition to the above quantitative analysis, we compare our selected pairwise optimisation method (MIRTk) with a broadly used learning-based method (Voxelmorph) and report quantitative results assessing the transformations' accuracy and regularity.

[10] Schnabel, J. A., Rueckert, D., Quist, M., Blackall, J. M., Castellano-Smith, A. D., Hartkens, T., ... & Hawkes, D. J. (2001). A generic framework for non-rigid registration based on non-uniform multi-level free-form deformations. In *Medical Image Computing and Computer-Assisted Intervention–MICCAI 2001: 4th International Conference Utrecht, The Netherlands, October 14–17, 2001 Proceedings 4* (pp. 573-581). Springer Berlin Heidelberg.

[11] Balakrishnan, G., Zhao, A., Sabuncu, M. R., Guttag, J., & Dalca, A. V. (2019). Voxelmorph: a learning framework for deformable medical image registration. *IEEE transactions on medical imaging*, 38(8), 1788-1800.

[12] Schuh, A., Qiu, H. & HeartFlow Research. deepali: Image, point set, and surface registration in PyTorch, DOI: 10.5281/zenodo.8170161

2. The data selection approach. Selecting 3000 samples for 6 subgroups may not be the best way of using the databank. For example, can the authors use all materials that fits the criteria of each subgroup? No data bias will be introduced as each subgroup is analysed independently.

Similar: “page 8: We randomly select 200 subjects with high liver fat.” Why not analyse ALL subjects with high liver fat (e.g. in male overweight group)?

Page 8: “We select 200 subjects with CAD, 200 subjects with diabetes and 200 subjects from the healthy atlas group.” The authors should use all subjects that are available and fit the criteria, to utilise the large data size and statistical power of UKBB.

We agree with the reviewer on this point, and we believe that incorporating all the data available makes better use of the capacity of the UK Biobank. We, therefore now incorporate all the available data and kindly refer to Table 1 for more detailed numbers.

3. Very limited report of results is the reviewer’s main concern.

- There are no validation results of the affine and non-rigid registration algorithms on UKBB. The authors should report dice coefficients and bland-altman plots on organ regions of interest (ROI), e.g. (by transforming the ROI masks back to original images,

and comparing them with original annotations) to provide more insight into the accuracy, bias and probability of failure/misregistration. This is evidently crucial, as the authors observed diffused noise and significant voxels introduced by mis-registration in Figs 5 and 6.

We would like to thank the reviewer for this comment. We agree that reporting quantitative metrics on the registration performance is beneficial for assessing our results. We have therefore included the above-mentioned metrics on registration accuracy and transformation regularity in the Results section (**Experimental setup**). These metrics measure structural overlap as well as the smoothness of the applied deformation. The results are summarised in Table 3, which contains the quantitative results of comparing the registration methods.

- Each result subsection only has one demonstrative example for unbiased atlas, with no summary of performance or findings on all samples.

We incorporated a visualisation of the unbiasing results for all samples and groups in the supplementary material in Figure 6. Moreover, we have included example visualisations for all groups for the final atlases (Figures 7 and 8) and the VBM analysis (Figures 9 and 10).

- The results section has a large volume of texts that describe the methods and approaches, and of discussion. They should be moved to the right sections.

We thank the reviewer for the suggestion. We restructured the content in order to increase the clarity and the readability of the manuscript. More specifically, we clearly separated methods from the results section by, for example, moving the description of the VBM to a dedicated paragraph in the method section. Furthermore, we renamed the previous “**Results**” section to “**Experiments and Results**” to reflect more accurately the content.

- Subsequently, the results section should report, e.g. new findings, association with outcomes, diagnostic performance of the derived atlas, the performance of label propagation. The authors mentioned label propagation in the discussion, this is straightforward to run and can significantly improve the paper.

As Reviewer #1 pointed out, the advent of large vision models such as Segment anything in medical images, Total Segmentator, make the segmentation part easier and renders label propagation less relevant.

Therefore, we have decided not to incorporate this into this manuscript, but rather strengthen our experiments on anomaly detection. We have extended our analysis of the application on CAD and diabetes to highlight differences in fatty tissue distribution in the abdomen. We have, for example, identified differences between these two patient groups regarding significant changes in fatty tissue. Here, we observe changes around

the pancreatic region only for the diabetic subgroup, aligning with medical research [8]. These specific findings have been confirmed by expert radiologists, assessing the medical validity of our results.

[8] Marshall, S. M. The pancreas in health and in diabetes. *Diabetologia*, 1962–1965 (2020)

Reviewer #1 (Remarks to the Author):

The authors have successfully incorporated the Reviewer's comments and made the work even better. No further concerns.

Response: We thank reviewer #1 for taking the time to provide insightful feedback on our work.

Reviewer #2 (Remarks to the Author):

The authors have addressed all my previous comments point by point. This is a much more strengthened manuscript and I have no further suggestions.

Response: We thank reviewer #2 for taking the time to provide valuable feedback on our work.